# Germline-Specific Repetitive Elements in Programmatically Eliminated Chromosomes of the Sea Lamprey (*Petromyzon marinus*)

**DOI:** 10.3390/genes10100832

**Published:** 2019-10-22

**Authors:** Vladimir A. Timoshevskiy, Nataliya Y. Timoshevskaya, Jeramiah J. Smith

**Affiliations:** Department of Biology, University of Kentucky, Lexington, KY 40506, USA; nti225@uky.edu (N.Y.T.); jjsmit3@uky.edu (J.J.S.)

**Keywords:** chromosome elimination, chromatin diminution, sea lamprey, chromosome lagging, repetitive DNA

## Abstract

The sea lamprey (*Petromyzon marinus*) is one of few vertebrate species known to reproducibly eliminate large fractions of its genome during normal embryonic development. This germline-specific DNA is lost in the form of large fragments, including entire chromosomes, and available evidence suggests that DNA elimination acts as a permanent silencing mechanism that prevents the somatic expression of a specific subset of “germline” genes. However, reconstruction of eliminated regions has proven to be challenging due to the complexity of the lamprey karyotype. We applied an integrative approach aimed at further characterization of the large-scale structure of eliminated segments, including: (1) in silico identification of germline-enriched repeats; (2) mapping the chromosomal location of specific repetitive sequences in germline metaphases; and (3) 3D DNA/DNA-hybridization to embryonic lagging anaphases, which permitted us to both verify the specificity of elements to physically eliminated chromosomes and characterize the subcellular organization of these elements during elimination. This approach resulted in the discovery of several repetitive elements that are found exclusively on the eliminated chromosomes, which subsequently permitted the identification of 12 individual chromosomes that are programmatically eliminated during early embryogenesis. The fidelity and specificity of these highly abundant sequences, their distinctive patterning in eliminated chromosomes, and subcellular localization in elimination anaphases suggest that these sequences might contribute to the specific targeting of chromosomes for elimination or possibly in molecular interactions that mediate their decelerated poleward movement in chromosome elimination anaphases, isolation into micronuclei and eventual degradation.

## 1. Introduction

The sea lamprey possesses a distinctive mechanism of differentiating somatic and germline lineages that is achieved by discarding large portions of the genome during early stages of development [1,2]. Similar largescale changes in genome content and structure have been described in several groups of phylogenetically diverse organisms [3] including: ciliates [4,5,6], nematodes [7,8], sciarid flies [9,10], copepods [11], chironomids [12], several hagfish species [13,14,15,16], songbirds [17,18], and at least two lamprey species [19,20]. These changes in genome content/structure are generally known as programmed genome rearrangement (PGR) [21,22]. The diversity of subcellular events associated with DNA elimination and the patchy taxonomic distribution of PGR reflect the repeated evolution of independent mechanisms regulating the reproducible targeting and processing of segments that are slated for elimination in somatic cell lineages. 

Despite the diverse origins of PGR within metazoan lineages, the loss of germline-specific segments shares common features across distantly related taxa. In most taxa, elimination appears to largely play out during the progression of specific early embryonic anaphases, wherein eliminated chromatin exhibits differential motion relative to somatically retained chromatin (e.g., as observed in nematodes, copepods, chironomids, sciarid flies, and lampreys). This general pattern can be divided into nonexclusive two categories: chromosome elimination per se that involves the removal of whole intact chromosomes (hagfish [23], sciarid flies [9], and song birds [17,18,24]) and chromatin diminution which includes steps of excision and rejoining, or telomere restoration (roundworms [25,26], copepods [27,28], chironomids [29], and some hagfish [30]). In lampreys, eliminated chromatin is ultimately packaged into discreet structures, termed micronuclei, wherein eliminated sequences accumulate DNA methylation marks and are ultimately degraded [2,20,31]. 

Programmed deletions have been reported to target chromosomal segments containing highly repetitive DNA which are often packaged into transcriptionally repressive heterochromatin prior to elimination [8,30,32,33,34]. More recent genome sequencing studies have revealed that programmed deletions result in the loss of hundreds of protein-coding genes in the sea lamprey [2,35], *Ascaris* [8] and zebra finch [30,36], many of which are thought to function in the maintenance of pluripotency, proliferation, development, and differentiation of germline tissues. These sequencing studies are consistent with previous observations in the Taiwanese hagfish *Paramixine sheni*, wherein it was reported that both C-band positive chromatin (presumptive heterochromatin) and C-band negative chromatin (presumptive euchromatin) are eliminated from somatic tissues [30]. Additionally, in sciarid flies, chromosome elimination contributed to both genomic differentiation of germline and soma, as well as sex determination, indicating that losses are likely to have specific genetic effects [10]. The presence of both protein-coding genes with presumptively critical functions and numerous high copy elements raises the questions as to whether repetitive elements themselves are truly junk targeted for elimination, passive passengers that are simply being carried along for the ride, or perhaps functionally relevant sequences that actively participate in the process of elimination.

In the present work, we focused on cytogenetically recognizable aspects of PGR in the sea lamprey. The sea lamprey’s somatic diploid karyotype consists of 168 small dot-like chromosomes [19,37,38]. Its germline karyotype was previously estimated to consist of ~99 small chromosomes indicating that lampreys might eliminate up to 15 entire chromosomes during PGR [19,39]. However, the complex morphology of germline metaphase spreads and the presence of numerous small chromosomes has thus far prevented a precise description of the germline karyotype. Previous studies showed that in lamprey, eliminated chromatin migrates more slowly during early embryonic anaphases (during the sixth through ninth embryonic cell divisions) and maintains contact with the original metaphase plane via an as-yet poorly understood mechanism [2,20]. We surmise that these interactions must be mediated by some specific feature(s) of the germline sequence that mediate their unique migration during specific embryonic metaphases; however, these features are currently unknown and specific candidates have yet to be identified. Moreover, it remains unclear the degree to which differences in chromosome number arise from wholescale loss of chromosomes vs. breakage and joining of remodeled segments. Here we use computational prediction of germline-enriched motifs (repeats) and FISH hybridization to more precisely characterize the content, structure and organization of eliminated chromosomes.

## 2. Materials and Methods 

### 2.1. Research Animals

Animals were obtained from Lake Michigan via the Great Lakes Fisheries Commission and maintained under University of Kentucky IACUC protocol number 2011-0848 (University of Kentucky Institutional Animal Care and Use Committee). For tissue sampling, animals were euthanized by immersion in buffered tricaine solution (1.0 g/L), dissected, and tissues for DNA isolation were immediately frozen. For meiotic chromosome preparations, testes were extracted from non-spermating males, collected temporally in PBS before processing. 

### 2.2. Production of Lamprey Embryos

In-vitro fertilizations were performed with sexually mature adult animals. Eggs and sperm were collected in crystallization dishes and incubated in 10% Holtfreter’s solution for 10 min to permit fertilization [40]. After visually confirming activation, embryos were rinsed in distilled water to remove excess sperm and maintained in 10% Holtfreter’s solution at 18 °C throughout development [41]. At days 1, 1.5, and 2 post-fertilization, live embryos were collected in 15-mL centrifuge tubes and fixed in MEMFA Fixative for 1 h, rinsed in PBS, dehydrated in increasing concentrations of methanol, and stored in methanol at -20 °C as previously described [42].

### 2.3. PACT Clearing

MEMFA fixed embryos were embedded in hydrogel and cleared according to a PACT (passive clarity technique) protocol optimized for lamprey embryos [2,43]. Prior to clearing, embryos were gradually rehydrated in PBS then perfused with hydrogel monomer solution (5% acrylamide supplemented with 0.5% VA-044) by incubating overnight at 4 °C. Hydrogel polymerization was performed at 37 °C for 2.5 h. After brief washes with PBS, embryos were transferred to a 50 mL screw-cup tube and incubated in stripping solution (8% SDS in PBS) for 5 days at 37 °C with gentle rotation. Upon reaching transparency, samples were washed in PBS with five buffer changes over the course of a day and transferred into staining solution (PBS, pH = 7.5, 0.1 Triton X-100, 0.01% sodium azide). Cleared embryos were stored at room temperature prior to downstream processing. 

### 2.4. Preparation of Metaphase Spreads 

Meiotic cell preparations were made from testes of non-spermating males. About 1 cm^3^ of tissue was homogenized in Dounce Grinder, the cells were treated with HEPES (0.01 M) buffered 0.075 M KCl hypotonic solution, pH = 7.4, and fixed with methanol:glacial acetic acid (3:2) fixative solution. Mitotic spreads were obtained from 16 dpf embryos using overnight exposure to colchicine (0.04%), homogenization, and buffered hypotonic treatment. Cell suspensions were applied to a clean steamed glass slide and immediately placed in a humidity chamber at 55 °C to facilitate proper chromosome spreading [44]. 

### 2.5. C_0_t DNA Isolation 

For highly repetitive DNA (C_0_t) isolation, genomic lamprey DNA was extracted by phenol-chloroform method [45], and repetitive fractions were isolated using S1 nuclease to digest single-stranded (low copy) DNA as described previously [46].

### 2.6. Computational Prediction of Germline-Specific Repeats

Abundant k-mers (k = 31) were identified from publicly available lamprey sperm (SRR5535435) and blood (SRR5535434) DNAseq datasets using Jellyfish version 2.2.4 [47]. Minimal copy-number thresholds for defining abundant k-mers were set at 3X the modal copy number: 165 for sperm and 180 for blood. Abundant k-mers were extracted and assembled into a set of high-identity repetitive elements using Velvet version 1.2.10 [48] with a hash length of 29. 

These de novo assembled repeats were aligned to repetitive sequences generated from the lamprey reference genome PIZI00000000.1 [35] by RepeatModeler [49]. Sequences aligning to RepeatModeler repeats with >90% identity over 80% of their length were replaced by the corresponding longer sequence. The previously characterized *Germ*1 repeat was also added to the set leading to the exclusion of 13 repeats that matched it with 99% identity.

An enrichment analysis was performed by separately aligning paired-end reads from sperm (SRR5535435) and blood (SRR5535434) DNAseq datasets to the set of assembled reference repeats using BWA MEM [50]. The DifCover pipeline [51] was used to calculate enrichment scores and stage 2 of the analysis pipeline was run with parameters that (1) prevent splitting sequences to intervals shorter than 5Kb (v = 5000); (2) report intervals of any length (l = 0); (3) assure reliable minimal coverage (a = b = 10) by either sperm or blood reads; and (4) allow bases with depth of coverage as high as the observed maximum of 11.8 M (A = B = 12 M). A set of 118,634 intervals generated by DifCover was filtered to identify 171 highly abundant and germline-specific sequences with enrichment scores of more than 5 and estimated span sizes of more than 40 Kb (Appendix A). The estimated genomic span of these repeats was computed as [length of sequence*(sperm coverage/modal sperm coverage)], where modal sperm coverage = 73.

Clustering of 171 highly abundant and germline-specific sequences was performed using CD-HIT-EST (v4.6, with parameters: -c0.8, -G0, -aS 0.3, -aL 0.3, -sc 1, -g 1, -b 4) [52], resulting in the identification of 30 clusters. We then cross aligned (blastn with -word_size 11) [53] sequences from separate clusters and found that some clusters could be further merged (required to have at least four hits), resulting in 20 clusters, 8 of which contained multiple sequences and 12 of which were singletons. For characterization of larger-scale repetitive structures, the genomic scaffolds with the largest number of hits to each germline-specific element were identified by BLAST alignment (blastn, word_size 11, at least 80% of bases aligned) (Appendix A).

### 2.7. Fluorescence in Situ Hybridization

Probes: To produce germline-specific probes, we used a DNA library that was generated from laser capture microdissected lagging anaphases as part of a previous study [20]. An aliquot of this library was amplified using the GenomePlex^®^ WGA Reamplification Kit (Millipore Sigma, Burlington, MA, USA) following the manufacturer instructions. Fluorescent laser-capture painting probe (LC probe) was generated using a modified version of the manufacturer’s protocol: Cyanine 3-dUTP (Enzo, Farmingdale, NY, USA) was used along with 10 mM dATP, dCTP, dGTP, and 3 mM dTTP, replacing dNTP manufacturer kit mix. Probes for the *Germ*1 repeat and C_0_t1-2 fraction of genomic DNA were produced using nick-translation of either an isolated BAC clone (*Germ1*) [19] or C_0_t fractions according to previously published protocol [45,54] using respectively Fluorescein-12-dUTP (Thermo Fisher Scientific, Waltham, MA, USA) and Cyanine 3-dUTP (Enzo, Farmingdale, NY, USA) as labeled nucleotides. The TelG-FAM PNA-probe (TTAGGG repeats, PNA Bio) was used as a telomere-specific probe. Probes for repetitive sequences were labeled via conventional PCR using a dNTP mixture containing 1 mM dATP, dCTP, dGTP, and 0.3 mM dTTP, and one fluorophore: Cyanine 3-dUTP (Enzo, Farmingdale, NY, USA), Cyanine 5-dUTP (Enzo, Farmingdale, NY, USA), Fluorescein-12-dUTP (Thermo Fisher Scientific, Waltham, MA, USA), or ChromaTide^®^ Alexa Fluor^®^ 488-5-dUTP (Thermo Fisher Scientific, Waltham, MA, USA). Each PCR amplification was performed using GoTaq^®^ DNA polymerase (Promega, Madison, WI, USA, 0.6 units/25 μL reaction), Colorless GoTaq^®^ Reaction Buffer, 1 μg of genomic DNA template, and 100 ng of oligonucleotide primer. PCR cycling conditions included a 3-min initial denaturation step at 95 °C, three-step thermal cycling consisting of a 30-s denaturation at 95 °C, a 30-s primer annealing step at 55 °C, and a 30-s extension step at 72 °C. A final extension at 72 °C was performed on all reactions to ensure the production of full-length amplicons. Depending on the nature of repeats, amplification reactions were performed over 34 (germline-specific repeats) or 30 (somatically-retained repeats) cycles. Amplification of somatically-retained repeats required fewer cycles in order to account for their higher copy numbers. 

DNA in situ hybridization on whole embryos and cytogenetic slides: 3-D FISH on whole embryos was performed according to a previously described protocol [2,20,55]. For individual hybridization experiments, four to five embryos were placed in a 2-mL tube and incubated in 8% sodium thiocyanate solution overnight at 37 °C, washed in PBS for 1 h, and placed in 50% formamide in 2xSSC for 2–3 h at 45 °C. For hybridization, 50% formamide/2xSSC solution was replaced with a 30-μL hybridization mix consisting of 50% formamide, 10% dextran sulfate, 0.01% sodium azide, and 250 ng of labeled DNA-probe. Embryos were pre-incubated overnight at 37 °C to permit penetration of probes, after which the probe and target DNA were denatured by heating samples to 75 °C for 5 min, chilled on ice for 2 min, and then moved to 37 °C for overnight hybridization. Three subsequent washes: 50% formamide in 2xSSC, 0.4x SSC, 0.3% Nonidet-P40 and 2xSSC, 0.1% Nonidet-P40 for 15 min each at 45 °C, were performed the following day. Embryos were placed on a slide in a drop of ProLong™ Gold Antifade Mountant with DAPI (Thermo Fisher Scientific, Waltham, MA, USA) and enclosed under a coverslip with slight pressure. 

FISH on chromosome preparations was carried out according to a standard protocol for chromosome spreads [56] with modifications [46]. For hybridizations involving the LC probe, a 20x excess of unlabeled C_0_t2 DNA was used to suppress hybridization to somatic repeats. After denaturation, each hybridization mixture was incubated at 37 °C for 30 min, then applied to chromosome spreads that had already been denatured and dehydrogenized [54]. Probes that were co-hybridized with LC were denatured separately and pooled with the LC probe after suppressive incubation, but before applying to the slides. 

C_0_t2-CGH: Repetitive fractions of genomic DNA were isolated from testes and liver DNA that were extracted from an adult male via phenol chloroform purification, as previously described [45]. A total of 1.5 μg testes C_0_t2 DNA was labeled with Cy3-dUTP (Enzo, Farmingdale, NY, USA) by nick-translation in a final volume of 50 μL; the same amount of liver C_0_t2 was labeled with fluorescein-12-dUTP (Thermo Fisher Scientific, Waltham, MA, USA). After labeling, probes were precipitated by adjusting the solution to 70% ethanol in the presence of 20 μg single-stranded sheared salmon sperm DNA (Sigma-Aldrich), followed by centrifugation at 14,100 G. The resulting pellet was air dried and resuspended in 50 μL of hybridization buffer. Ten μL of each probe and 10 μL of hybridization buffer were mixed prior to hybridization to whole embryos. Conditions for hybridization were as described above. Hybridizations were carried out on embryos that were actively undergoing elimination at the time of fixation (1.5–2 days post fertilization: dpf). Fluorescence intensity was measured using images of cells containing micronuclei, and mean integrated fluorescence density of primary nuclei was used as background fluorescence respective to micronuclei.

Microscopy and image analysis: After FISH and immunostaining, slides were analyzed with an Olympus-BX53 microscope using filter sets for DAPI, FITC, TexasRed, and Cy5. Images were captured using CellSence software (Olympus, Shinjuku, Tokyo, Japan). For thicker samples, such as intact cells from PACT cleared embryos, the extended focal imaging function was used in order to generate a single deep-focus image. Images from each filter set were captured separately, and merging of channels was performed using Adobe Photoshop CC 2017. Measurements of fluorescence intensity were carried out in ImageJ 1.48v (NIH) using raw images and options “integrated density” and “mean gray value” for selected areas on captured images.

## 3. Results and Discussion

### 3.1. Comparative Hybridization Indicates the Presence of Germline-Specific Repeats

To assess whether eliminated chromatin was likely to be enriched with germline-specific repeats, we performed comparative hybridization of repetitive DNA (C_0_t2 fractions) from somatic (liver) and germline (testes) genomic DNA within intact embryos 1.5–2 days post fertilization: dpf [20,43]. Cells containing micronuclei were used for comparative analysis of germline/soma repetitive content (Figure 1A). Micronuclei were found to be highly enriched in germline-derived repetitive DNA (*p* < 0.0001, *n* = 33, DF = 32) (Figure 1B). A similar enrichment was also observed in the analysis of ~30 eliminating anaphases: Cy3-labeled germline repeats exhibited substantially higher fluorescence intensity within lagging chromatin than FITC-labeled somatic repeats (Figure 1C and Appendix A). In elimination anaphases, somatically-retained repeats were observed to hybridize primarily with pericentromeric and peritelomeric regions forming dot-shaped signals and were largely absent from the internal regions of lagging chromosomes (Appendix A, see also Appendix A). This is interpreted as evidence that germline-specific and somatically-retained chromosomes share a similar complement of pericentromeric repeats and that pericentromeric repeats account for most of the rapidly annealing fraction of somatic DNA, whereas germline-specific repeats account for much of the rapidly annealing fraction of germline DNA. In conjunction with the observation that the centromeres of eliminated chromosomes exhibit poleward motion during elimination anaphases, this seemingly indicates that both eliminated and retained centromeres retain a capacity to form functional kinetochores at least to some extent [2]. 

Previous studies have shown that a majority of micronuclei in 1–3 dpf sea lamprey embryos contain the germline-enriched repeat *Germ1* [2]. In order to determine whether germline C_0_t2 hybridization patterns could be explained simply by the presence of *Germ*1, we co-hybridized a *Germ*1-specific probe with labeled C_0_t1 DNA to elimination anaphases (Figure 1C). Owing to its highly repetitive nature, labeled C_0_t1 DNA hybridizes to centromeric sequences and a small subset of subtelomeric repeats. These hybridizations reveal that *Germ*1 localizes to discreet regions within eliminated chromosomes, primarily near the centromeric ends of these acrocentric chromosomes, with the remaining portions of eliminated chromosomes being largely devoid of *Germ1*. Consistent with these observations, *Germ*1 probes were also found to hybridize to a subset of micronuclei (68% in D2 embryos and 66% in D2.5 embryos), whereas germline C_0_t2 DNA (repeats) hybridized to all micronuclei in D2 and D2.5 dpf embryos. These observations were taken as evidence that repetitive elements in addition to *Germ*1 were likely to be enriched within germline-specific chromatin, present in regions outside of the defined *Germ*1 clusters and, in some cases, fragmented from *Germ*1-positive regions prior to micronucleus formation. 

### 3.2. Computational Identification of Germline-Specific and Centromeric Repeats

Consistent with the above findings, previous computational analyses of the repetitive content of germline and somatic DNA indicated that there is likely a large number of distinct repetitive elements that are unique to the germline [35]. To generate more precise consensus assemblies of individual repeat families and relative copy number estimates for each family, we performed a *de novo* assembly of repetitive elements that were seeded from a complete list of 31 mers that were abundant in sperm and/or blood reads. This assembly yielded a total of 130,632 consensus repetitive sequences. These sequences were merged with repeats that were identified within genomic scaffolds via RepeatModeler and an updated sequence of the *Germ*1 [49], yielding a total of 119,842 model repeats. Copy number estimates for consensus repeats were generated by remapping sequencing data from sperm and blood to obtain separate metrics of sequence coverage. A majority of repetitive sequences were abundant in both sperm and blood, however, the distribution of coverage ratios contains a notable tail corresponding to germline-enriched sequences (Figure 2 and Appendix A). 

All predicted high-copy elements with enrichment scores [log2(standardized sperm coverage/blood coverage)] exceeding 5 and an estimated span exceeding 40 kb, when summed across all copies, were extracted for downstream analysis (Appendix A). Subsequent inspection of these 171 predicted elements revealed similarities among subgroups of repeats, and semi-automated clustering revealed that these high-copy repeats could be grouped into 20 distinct clusters, i.e., repeat families. The representatives of six clusters, with a combined span size of more than 500Kb, were designated *Germ*2–7 (Figure 2 and Appendix A).

Examination of the sequences of *Germ*2–7 and genomics scaffolds PIZI00000000.1 [35] containing these repeats revealed that all of these high-copy germline-specific repeats occur as tandem arrays. Each repetitive element appears to contain a short (13–57 bp) somewhat conserved core sequence (Appendix A). Tandem arrays of these core sequences are frequently disrupted by small insertions or deletions and, at larger scales, cassettes of tandem repeats are further duplicated as inverted repeats. Homology searches against the current Repbase release (09-26-2019), using RepeatMasker [49], revealed no known homologs for any of these satellite repeats.

Arbitrarily chosen representatives of each cluster were selected for primer design and PCR validation. Amplicons generated from these primers yielded a continuous range of fragment sizes (smear), as would be expected for primers designed within tandem repeats, and relative specificity to the germline (Appendix A). These same amplicons were used to generate probes (*Germ*2–7) for subsequent FISH analyses.

We previously used the rapidly annealing fraction of genomic DNA (C_0_t 1 or C_0_t2) to label high abundance satellite repeats [2,57]. These yielded strong signals on the poleward migrating ends of retained and lagging anaphase chromosomes (presumptive centromeres) and hybridized to the centromeric regions of metaphase chromosomes. However, these probes are not defined by a specific known sequence. In order to identify a more precise sequence with which to label centromeres, we selected repeats with the highest coverage in somatic shotgun sequence data (Appendix A, Appendix A). FISH revealed that the repeat with the highest somatic coverage, Pm-rep1, hybridized specifically to the centromeric regions in a pattern that is nearly identical to that of the rapidly annealing fraction (Appendix A). The Pm-rep1 probe also yields fainter signals near the telomeres (Appendix A), possibly due to cross-hybridization with subtelomeric repeats. The second-most abundant, Pm-rep2, yielded punctate signals spread across several chromosomes, and Pm-rep3 hybridized to a subset of ~28 chromosomes (12 of which produced bright signals).

### 3.3. Defining the Chromosomal Localization of Germline-Restricted Repeats

The lamprey karyotype is characterized by a large number of small chromosomes, a feature that presents significant challenges in the identification and characterization of individual homologs. Because repeats were predicted to be highly enriched in germline, we performed hybridizations against meiotic metaphases (spermatogenesis metaphase I) in order to characterize the distribution and location of these repetitive elements across lamprey chromosomes. Germline-specific chromosomes were identified on the basis of hybridization to a previously described laser-capture painting probe [20]. Additionally, we used two probes, a telomere-specific repeat (PNA probe) and centromere markers (labeled C_0_t1 fraction of genomic DNA from somatic lineage) to aid defining the bounds of individual chromosomes (Appendix A). 

These hybridizations yielded counts of 84 somatically-retained and 12 germline-specific chromosomes. In meiotic spreads, the LC probe hybridized with several entire chromosomes and a portion of one somatically-retained chromosomes that has been previously shown to hybridize to *Germ*1 in somatic metaphases (Appendix A). In total, counts of germline-specific (LC positive) and somatically-retained chromosomes yield a haploid number of *n* = 96 chromosomes in the lamprey germline, which is fewer than the previously estimated *n* = 99 [19,39].

In an attempt to uniquely identify individual germline-specific chromosomes, our six new computationally-predicted germline-specific repetitive elements (*Germ*2–7) and *Germ*1 were hybridized to metaphase chromosomes in three successive rounds of hybridization, which allowed us to localize all seven elements and the germline LC probe on the same set of meiotic metaphases. We analyzed at least 40 meiotic metaphase spreads and found that all six of the predicted germline-specific repeats hybridized exclusively to chromosomes that were marked by the germline LC probe (Figure 3). These hybridization patterns allowed us to verify that these elements are restricted to eliminated chromosomes and provided a means of individually identifying all 12 eliminated chromosomes. 

These hybridizations were used, in conjunction with reverse DAPI staining patterns, to develop an idiogram for all germline-restricted chromosomes and a map of the locations of *Germ*1–7 repeats on each chromosome (Figure 4). Remarkably, three repeats (*Germ* 1, 2, and 6) were found to be present on each of the 12 eliminated chromosomes, albeit with distinct distributions. *Germ*1 is present as dense signals that are located adjacent to the pericentromeric regions of all 12 eliminated chromosomes. This pattern contrasts with those of *Germ*2, which is typically located closer to the telomere, and *Germ*6 which generally shows a more diffuse patterns across the length of chromosomal arms. Unlike *Germ*1, the elements *Germ*2 and *Germ*6 do not cross hybridize with somatic chromosomes. The other four germline-specific repeats vary more broadly in their distributions across chromosomes and individual elements appear to be completely absent from one or more chromosomes (Figure 3). 

Over the course of examining spreads of testes meiotic metaphase-I chromosomes, it was noted that germline-specific chromosomes were generally clustered within meiotic spreads. Closer examination of these clusters revealed that groups of germline-specific chromosomes frequently contacted one another near their telomeres, forming structures reminiscent of meiotic chains (Appendix A). Similar meiotic chains or ring formations have been described in evening primrose [58], *Incisitermes schwarzi* termites [59], *Leptodactylus* frogs [60] and monotremes: platypus and echidnas sex chromosomes [61,62,63]. In these species, the formation of multivalent chains is thought to be associated with chromosomal translocations, as has been observed in mice with Robertsonian exchanges [64,65]. However, in the sea lamprey meioses, the number and content of chromosomes in each chain appears to vary from cell to cell (Appendix A). We speculate that the formation of these variable chains during meiosis might reflect a general tendency for germline chromosomes to interact with one another at their distal ends, foreshadowing stronger interactions that occur during elimination (see below). Meiotic chains are known to form when telomere adjacent elements share homology [61]. Superficially similar chains can also form telomere–telomere fusions in an apparently homology-independent manner, resulting from changes in telomere protein content, as has been observed in *Drosophila* telomere capping protein mutants [66,67]. While it seems unlikely that chained meiotic chromosomes in lamprey are the result of sporadic telomere–telomere fusions, due to the fact that they appear to be present in a majority of meiotic cells, we reason that these interactions could result from the fact that they share long stretches of germline-specific satellite sequences. It seems plausible that these sequences could also contribute to stronger interactions that are observed during anaphase, perhaps in addition to other protein and RNA factors. 

### 3.4. Chromosomal Localization of Germline-Restricted Repeats in Elimination Anaphases 

In order to directly resolve the spatial organization of germline-specific repeats during elimination, we performed 3D in situ hybridizations within fixed and PACT-cleared embryos (Figure 5). As expected from the analyses of meiotic spreads, *Germ*2–7 signals were visible only in lagging (eliminated) chromatin and were absent from retained somatic chromosomes (Figure 3 and Figure 5). Each of these repeats marks a distinct subregion of eliminated chromatin that corresponds to its relative position in meiotic metaphases. During anaphase, centromeric ends of eliminated chromosomes (marked by Pm-rep1 and adjacent *Germ*1 signals) are oriented toward the spindle poles (Figure 1C; Figure 6A, Appendix A). These hybridizations also revealed that germline-restricted repeats are generally arranged in an antiparallel pattern along the axis of anaphase elongation, suggesting altered migration involves tethering of germline-specific sister chromatids at their telomeres or subtelomeric regions (Figure 6A). The distal portions of the eliminated chromosomes remain near the metaphase equatorial plane with their proximal ends oriented toward the somatically-retained chromosomes, which results in a stretched, bridge-like morphology of lagging chromatin (Figure 6).

It is also worth noting that the sites of contact between bridging chromatids are generally characterized by a condensed morphology resembling heterochromatic blocks. It was previously demonstrated that these sites are enriched with repetitive DNA [2]. 3D DNA/DNA-hybridization of Germline-specific probes to embryonic lagging anaphases revealed that two elements (*Germ*2 and *Germ*4) frequently localize to the midline of lagging anaphases (Figure 5A,B and Figure 7), suggesting the possibility that these repeats may mark a domain of interaction between the subtelomeric regions of several chromosomes that possess dense *Germ*2 and *Germ*4 domains near their distal telomeres. To examine whether stretched lagging chromosomes retain intact telomeres on both arms, we carried out FISH using PNA (Peptide Nucleic Acid) probes to the vertebrate telomere repeat consensus sequence. These hybridizations reveal that stretched lagging chromatin possesses telomere signals on both ends, consistent with the idea that eliminated DNA is composed largely of entire germline-specific chromosomes (Figure 7 and Appendix A). In general, it appears that poleward-oriented (centromeric) telomere signals are significantly larger and brighter than their equatorially-oriented counterparts, which occasionally yield diffuse signals that are indistinguishable from background autofluorescence (Appendix A; Appendix A). It is tempting to speculate that lagging chromosomes are establishing telomeric contacts due to telomere shortening and the formation of adhesive ends. However, two observations speak against this simplistic interpretation. First, similarly variable hybridization intensities are also observed in meiotic and mitotic metaphases (i.e., non-eliminating divisions; Appendix A; Appendix A). Second, medially-oriented telomere signals often appear as distinct pairs of signals within regions that are overlain by broader hybridization signals from repetitive DNA (i.e., *Germ*2, Figure 6 and Appendix A). These observations suggest that interactions between lagging sister chromosomes may involve repetitive (e.g., *Germ*2, 4, 7) or other sequences near the telomere ends, although such interactions are not necessarily exclusive with telomere shortening in bridged chromatin.

## 4. Conclusions

The identification germline-specific repetitive elements and localization of these elements to meiotic and eliminating cells sheds new light on the number and structure of eliminated chromosomes and might be useful for further investigation of their behavior and sub-cellular orientation during elimination. Taken together, our observations suggest that interactions between eliminated chromosomes result in differential migration and eventual loss of large segments of the genome and are consistent with the loss of 12 entire germline-specific chromosomes. These observations do not preclude the occurrence of other smaller-scale elimination events, but suggest that these 12 germline-specific chromosomes contain a majority of the eliminated DNA. Analyses of the distribution of repetitive sequences across micronuclei (wherein lagging material is isolated prior to degradation) further indicate that some fragmentation likely takes place during the early stages of elimination, resulting in the formation of micronuclei that contain germline-specific DNA, but lack *Germ*1 (*Germ*1 cassettes are present on all eliminated chromosomes described here). This builds on earlier work demonstrating that early interactions and micronuclei formation events are followed by the accumulation of methylation marks and degradation of eliminated DNA within micronuclei [2].

These analyses highlight the highly regulated nature of PGR in lamprey and suggest an essential role of germline-specific repetitive elements in targeting large DNA segments (chromosomes) for elimination. We anticipate that further analyses of these repeats and their associated binding partners in eliminating cells will provide new insights into mechanisms that mediate chromosome segregation and the maintenance of genomic integrity during mitosis. 

Over the longer term, we envision that the precisely targeted mechanisms underlying PGR might ultimately be coopted as a means of experimentally manipulating the behavior of specific intact chromosomes during anaphase. Our analyses demonstrate that lagging chromosomes likely have all fundamental morphological attributes necessary for normal chromosome segregation: Their centromere ends orient toward the spindle poles; the chromatin itself is oriented in line with the spindle filaments, implying interactions with spindle motor proteins; and lagging chromosomes retain intact telomeres. Moreover, they appear to segregate normally during the first several embryonic cell divisions and in the germline [2]. We anticipate that further work aimed at understanding factors that interact with germline-specific sequences, including repetitive elements [68], will shed light on the cellular mechanisms that identify germline-specific sequences and mediate their differential motion during anaphase.

## Figures and Tables

**Figure 1 genes-10-00832-f001:**
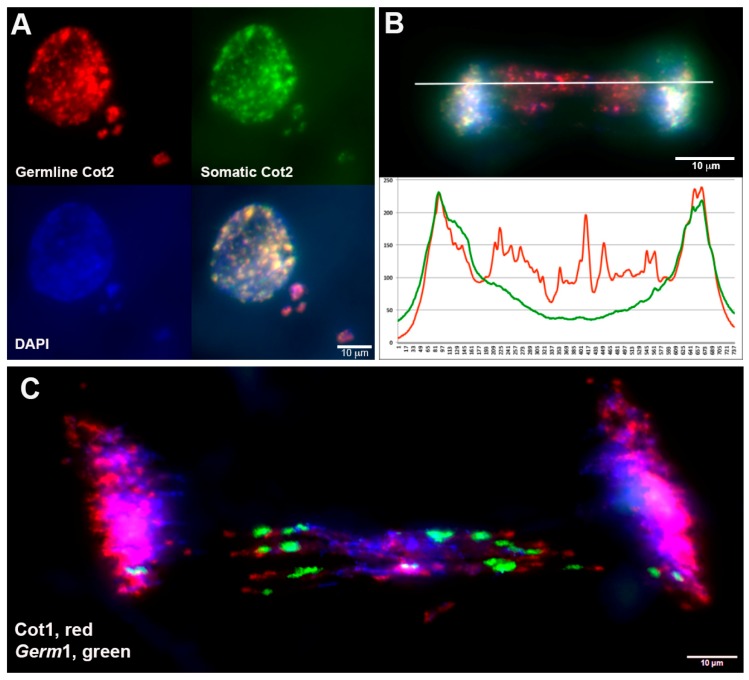
Repetitive DNA in situ hybridization in eliminating cells of 1.5 dpf sea lamprey embryos. (**A**) An example of competitive hybridization repetitive DNA fraction from testes (red) and liver (green) in a cell containing micronuclei. (**B**) Hybridization of C_0_t2 DNA-probes from germline (red) and somatic (green) to a lagging anaphase and a corresponding fluorescence intensity profile (measurement plane is marked by a horizontal line). (**C**) Hybridization of C_0_t1 (red) and *Germ*1 (green) to a lagging anaphase.

**Figure 2 genes-10-00832-f002:**
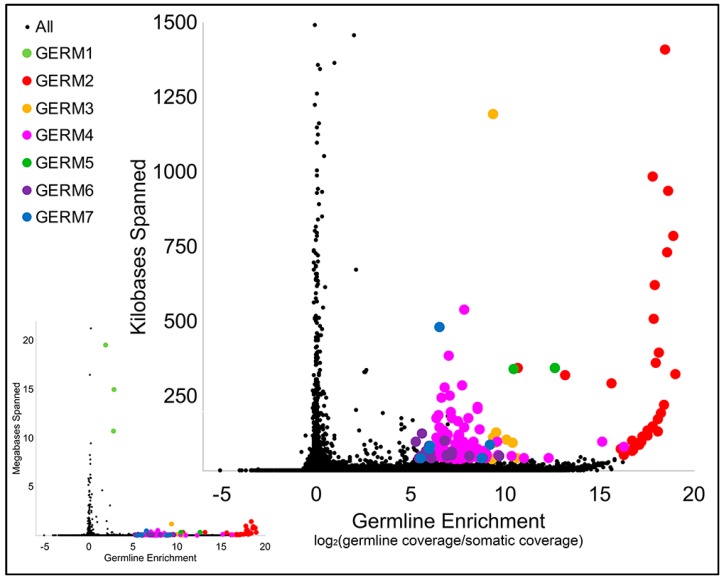
Estimates of the total genomic span and germline enrichment of repetitive sequences in the lamprey genome. The plot on the lower left shows enrichment values for the entire collection of reconstructed repetitive elements, including *Germ1*. The larger plot on the upper right focuses on repeats that span <1.5 megabases. Sequences belonging to repeat classes selected for hybridization are highlighted by colors corresponding to their repeat class.

**Figure 3 genes-10-00832-f003:**
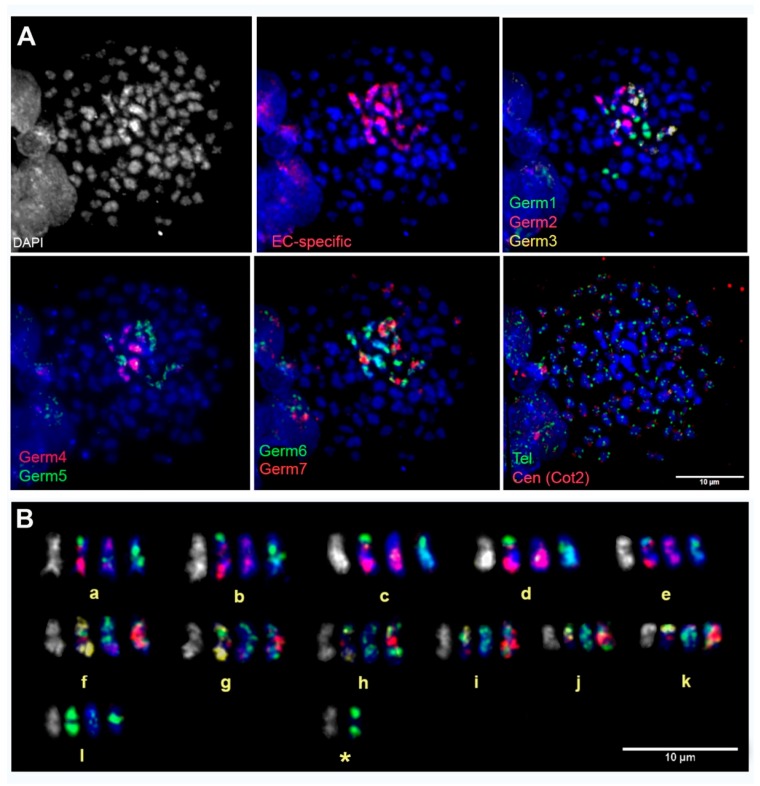
Chromosomal localization of the germline-restricted repeats. (**A**) FISH of *Germ*1–7, repeats and telomeric/centromeric (liver C_0_t2) probes on spermatid spreads, with DAPI counterstaining (grey or blue). (**B**) A karyogram of germline-restricted chromosomes (including a somatically-retained bivalent that cross-hybridizes with *Germ1*: labeled with an asterisk). This bivalent presumably encodes somatic ribosomal RNAs, which share sequence homology with *Germ*1 [19]. Each chromosome in the karyogram (**B**) is shown in four states: 1: grayscale DAPI counterstain; 2: hybridized with *Germ*1 (FITC, green), *Germ*2 (Cy3, red), and *Germ* 3 (Cy5, pseudocolored in yellow); 3: hybridized with *Germ*4 (FITC, green) and *Germ*5 (Cy3, red); 4: hybridized with *Germ*6 (FITC, green) and *Germ*7 (Cy3, red).

**Figure 4 genes-10-00832-f004:**
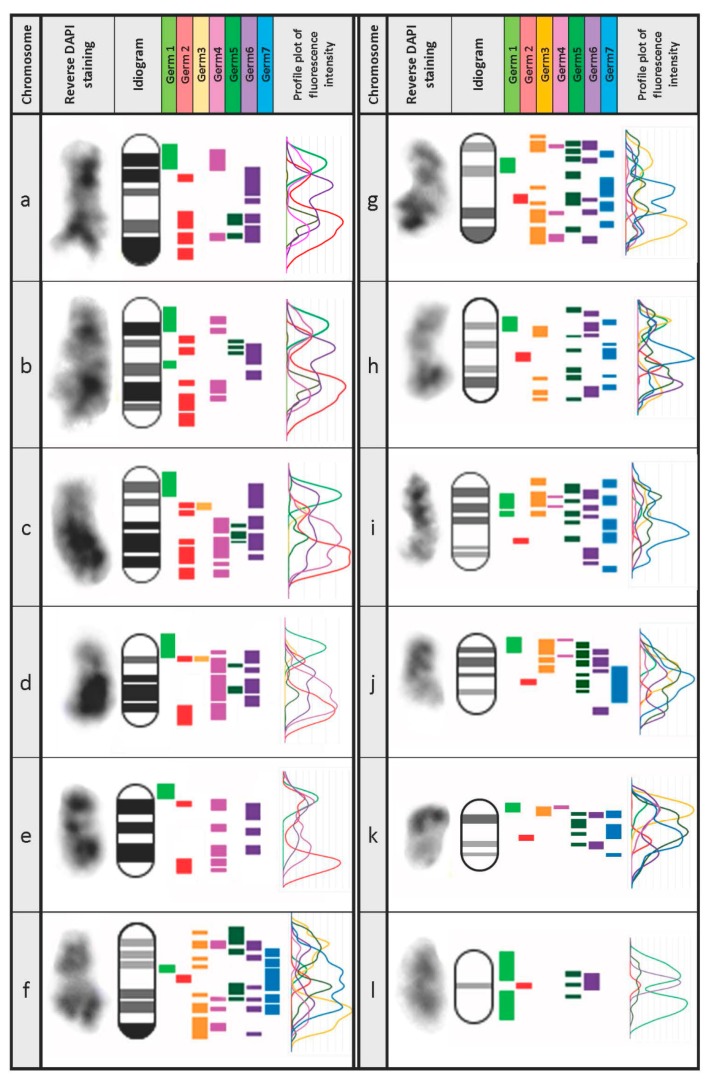
An idiogram of germline-specific chromosomes. Twelve germline-specific chromosomes (**a**–**l**) can be distinguished by DAPI staining and hybridization patterns of *Germ*1–7 repetitive elements in meiotic metaphase-I spreads (Figure 4). Profile plots were generated based on the fluorescence intensity of hybridized DNA-probes corresponding to each repeat.

**Figure 5 genes-10-00832-f005:**
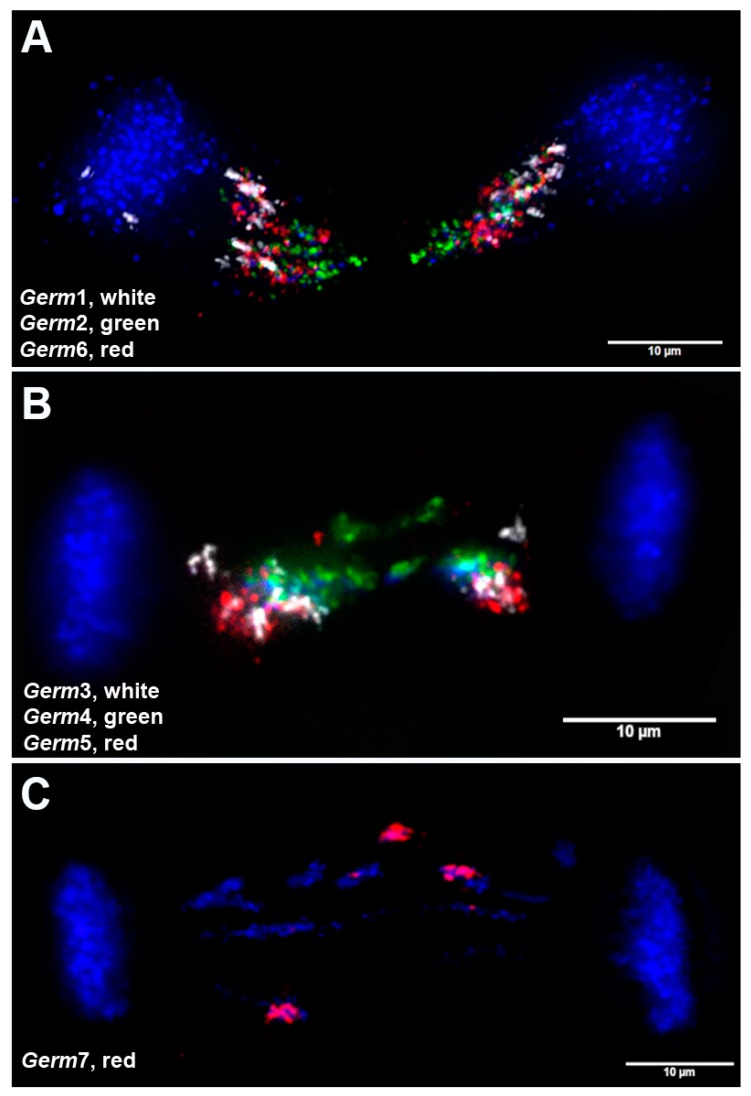
Lagging chromosomes and germline-restricted repeats. Representative examples from FISH of seven repetitive elements to lagging anaphases from 1.5 dpf sea lamprey embryos: (**A**) *Germ1* (white), *Germ2* (green), *Germ6* (red); (**B**) *Germ5* (red), *Germ4* (green), *Germ3* (white); (**C**) *Germ7* (red).

**Figure 6 genes-10-00832-f006:**
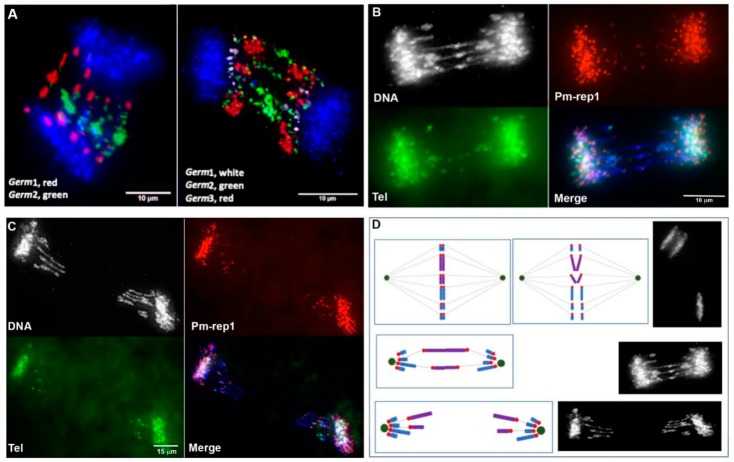
Chromosome lagging in anaphases from 1.5. dpf embryos, illustrating progression through anaphase and the antiparallel orientation of lagging chromosomes. (**A**) Earlier stages of anaphase chromosome separation show symmetrical hybridization patterns for germline-specific repeats. The repeat Germ1 is primarily located on the poleward ends of stretched chromosomes, which is consistent with its pericentromeric location. In contrast, the repeat Germ2 is localized to the midzone of bridging anaphases. (**B**,**C**) FISH of a probe for the pericentromeric repeat Pm-rep1 (red), telomere PNA probe (green), and testes genomic DNA (Cy5, shown in blue). The Pm-rep1 probe also yields fainter signals on the distal edges of lagging chromosomes which are often colocalized with distal telomeric signals. Chromatid contacts are characterized by denser DNA staining and FISH signals marking the telomeres/subtelomeres of sister chromatids which are often visible adjacent to each other (especially on panel **A**). Later in anaphase, (**C**) germline-specific chromosomes retain a stretched morphology and generally bear FISH signals from probes marking their edges (centromeres and telomeres). (**D**) Schematic depiction of chromosome elimination in the sea lamprey referencing features of the examples provided in panels **B** and **C**.

**Figure 7 genes-10-00832-f007:**
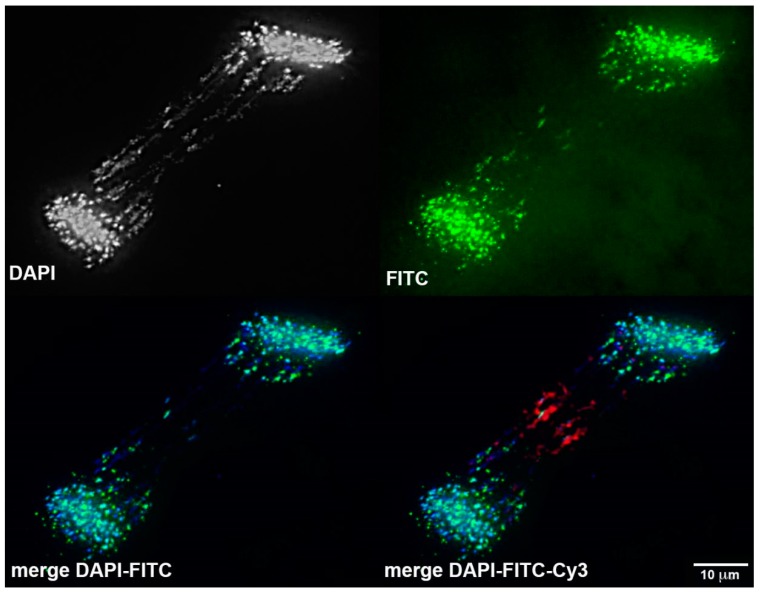
Telomeric contacts during chromosome elimination. Fluorescence in situ hybridization of telomere-specific (green) and Germ2 (red) to an anaphase from a 1.5 dpf embryo. See Appendix A for additional examples.

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
