# Peer review of "Germline-Specific Repetitive Elements in Programmatically Eliminated Chromosomes of the Sea Lamprey (Petromyzon marinus)"

_genes, 2019, doi:10.3390/genes10100832_

Round 1
Reviewer 1 Report
This is an excellent manuscript that I thoroughly enjoyed reading. The quality of the work and the presentation is high. I have only a few minor comments and suggestions of how the work might be improved:
It would be helpful to explain what micronuclei are for those who are not familiar with them. Are Germ2-7 specific to sea lamprey? A bit more discussion of what types of repetitive element these are would be useful. Fig. 2 is missing from the PDF.Author Response
We thank the reviewers for their insightful comments! We agree with all made comments and made several changes to the manuscript to incorporate these comments.
Reviewer #1:
It would be helpful to explain what micronuclei are for those who are not familiar with them. Are Germ2-7 specific to sea lamprey? A bit more discussion of what types of repetitive element these are would be useful. Fig. 2 is missing from the PDF.
We included additional and missing information in the manuscript, to better highlight the fact that these are satellite repeats with no known homology to repeats from other species represented in Repbase (lines 316-317).
Figure 2 was inserted in the manuscript body.

Reviewer 2 Report
The manuscript by Timoshevskiy et al. addresses a very difficult aspect of the sea lamprey biology: discovery and analysis of germline-specific repetitive sequences in eliminated chromosomes. After the computational identification of germline-specific repeats, they apply molecular cytogenetics methods to validate the findings and characterize the organization of these repeats during the elimination process. Additionally, they provide an ideogram representation of the 12 germline-specific chromosomes along with the map of the locations of the gernline-specific repeats. The findings are very interesting and worth publishing in Genes. However, there are several revisions/additions and text edits that I would like to see addressed.
Major revisions
1. Figure 2 is missing
2. Probes.
The description of the probes is a bit confusing. I thought you used only one fluorescent probe LC for the confirmation of germline specificity. In the text you describe: “Fluorescent LC probes were generated using a modified version of the 153 manufacturer’s protocol: Cyanine 3-dUTP (Enzo), ChromaTide® Alexa Fluor® 594-5-dUTP 154 (Thermo), or ChromaTide® Alexa Fluor® 488-5-dUTP…”. Which fluorescent LC probe was labeled with which of these fluorochromes?
Same here: “Probes for the Germ1 repeat 156 and C0t1-2 fraction of genomic DNA were produced using nick-translation of either an isolated BAC 157 clone (Germ1) [19] or C0t fractions according to previously published protocol [44,53] using Cyanine 158 3-dUTP (Enzo), Cyanine 5-dUTP (Enzo), or Fluorescein-12-dUTP (Thermo) as labeled nucleotides.” Is Germ1 labeled with Cyanine 158 3-dUTP (Enzo), Cyanine 5-dUTP (Enzo), or Fluorescein-12-dUTP? Is C0t1 labeled with Cyanine 158 3-dUTP (Enzo), Cyanine 5-dUTP (Enzo), or Fluorescein-12-dUTP?
Please provide which label and PCR protocol you used for each of the Germ2-7 and PM_rep1 repeats as well. The PCR protocol can be provided in Table S3 and S4.
3. Image analysis.
Why do you need to use Photoshop for signal merging. Can you use CellSence software for this purpose instead?
What is the scale bar in Fig. 1B?
4. Results and Discussion.
How many embryos and micronuclei were analyzed to provide the evidence for their enrichment in germline-derived repeats? Please provide this information in the text.
Where does the assembly PIZI00000000.1 come from? Could you provide the reference if it is already published?
Which repeats have the highest somatic coverage? The column “Repeats with highest somatic coverage” in Table S1 is empty.
Minor revisions:
Page 1, Line 14: remove one space between “karyotype. We”
Page 1, Line 18: remove “of”
Page 1, Line 20: remove one space between “elimination. This”
Page 1, Line 35: remove “Smith, et al.”, problem with the citation formatting
Page 1, Line 36-38: the font formatting problem
Page 2, Line 1: “metazoan” and not “Metazoan”
Page 2, Line 55: remove one space between “Ascaris. [8]”
Page 2, Line 58: comma missing before “wherein”
Page 2, Line 60: remove one space between “tissues. These”
Page 2, Line 69-70: no comma before and after “indicating”
Page 2, Line 73: no comma before and after “in lamprey”
Page 3, Line 95: remove “fixed were”
Page 3, Line 96: problem with Celsius degrees’ formatting here and thereafter, e.g. see page 4
Page 3, Line 103: space missing between “8%” and “SDS”
Page 3, Line 130: the font formatting problem
Page 3, Line 130: add “publicly available” after “from”
Page 4, Line 139: bracket missing after coverage
Page 4, Line 155: “Thermo Fisher”?
Page 4, Line 180: spell “LC” for the first time
Page 4, Line 180: “was” is missing before “used”
Page 4, Line 181-184: the font formatting problem
Page 5, Line 194: spell “dpf” out for the first time and use the abbreviation throughout the manuscript, “dpf” and “dfp” is used inconsistently throughout the text
Figure 1. caption: “comparative” and not “competitive:
Page 6, Line 233: “contains” and not “contain”
Page 7, Line 248: “is” and not “are”
Page 7, Line 258: “S1 Table” or “Table S1”, the citation of supplementary tables is inconsistent throughout the manuscript
Page 8, Line 307: “retained chromosomes” and not “retained chromosome”
Page 10, Line 347: “appear” and not “appears”
Page 10, Line 353: use “,” instead of “:”
Figure 6. Caption: “dpf” and not “d.f.p”
Page 14, Line 386: sometimes you use “germline-specific repeats” and sometimes “germline specific repeat”. Please fix that for the consistency throughout the text.
Page 14, Line 412: wrong formatting of the text after “lagging”
Page 14, Line 401: “germline“ and not “Germline”
Page 15: This should be Figure 7.
Page 15, Line 418: the verb missing
Page 15, Line 426: “shed” and not “sheds”
Page 15, Line 430: “12” and not “twelve”
Page 15, Line 427: “investigation of”
Figure S4B is not well explained
Figure S5: “LC” should be spelled out in the caption
Figure S6: which one is the additional panel?
Table S1: The calculation description of the enrichment scores and the genomic span should be included in the table caption instead of the column headers
Table S2: What is CDHIT-cluster?
Author Response
We thank the reviewers for their insightful comments! We agree with all made comments and made several changes to the manuscript to incorporate these comments.
Responses to Reviewer's questions
Reviewer #2:
Major revisions
Figure 2 is missingFigure 2 was inserted in the manuscript body. We think it was deleted from original file during submission.
Probes.The description of the probes is a bit confusing. I thought you used only one fluorescent probe LC for the confirmation of germline specificity. In the text you describe: “Fluorescent LC probes were generated using a modified version of the 153 manufacturer’s protocol: Cyanine 3-dUTP (Enzo), ChromaTide® Alexa Fluor® 594-5-dUTP 154 (Thermo), or ChromaTide® Alexa Fluor® 488-5-dUTP…”. Which fluorescent LC probe was labeled with which of these fluorochromes?
Same here: “Probes for the Germ1 repeat 156 and C0t1-2 fraction of genomic DNA were produced using nick-translation of either an isolated BAC 157 clone (Germ1) [19] or C0t fractions according to previously published protocol [44,53] using Cyanine 158 3-dUTP (Enzo), Cyanine 5-dUTP (Enzo), or Fluorescein-12-dUTP (Thermo) as labeled nucleotides.” Is Germ1 labeled with Cyanine 158 3-dUTP (Enzo), Cyanine 5-dUTP (Enzo), or Fluorescein-12-dUTP? Is C0t1 labeled with Cyanine 158 3-dUTP (Enzo), Cyanine 5-dUTP (Enzo), or Fluorescein-12-dUTP?
Please provide which label and PCR protocol you used for each of the Germ2-7 and PM_rep1 repeats as well. The PCR protocol can be provided in Table S3 and S4.
We incorporated description of labeled probes in the manuscript text and added additional information to S3 and S4 Tables.
Image analysis.
Why do you need to use Photoshop for signal merging? Can you use CellSence software for this purpose instead?
It’s true that CellSence has the function of automatic capture and merging different channels. We elected to manually merge separate channels because this approach seems to provide more accurate adjustments of brightness for signals of varying color/intensity, and is less prone to artefactually overexposing certain spots.
What is the scale bar in Fig. 1B?
We have added the label (10 um).
Results and Discussion.How many embryos and micronuclei were analyzed to provide the evidence for their enrichment in germline-derived repeats? Please provide this information in the text.
The number of analyzed for fluorescence intensity MNi was 33 (DF=32). We incorporated this information in manuscript. Other analyses are more difficult to assess, because they involved squashing dissociated cell preps derived from several embryos under a coverslip.
Where does the assembly PIZI00000000.1 come from? Could you provide the reference if it is already published?
Reference was provided.
Which repeats have the highest somatic coverage? The column “Repeats with highest somatic coverage” in Table S1 is empty.
We apologize for this confusion. The mentioned column contains only 4 items (for Pm-rep1-3), which we now realize are essentially invisible in the table as it contains ~185K. We have now sorted those repeats to the top of the table so that they are easier to find.
Minor revisions:
We greatly appreciate for this section of review and made all recommended corrections.
Figure S6: which one is the additional panel?
We made comment in the capture.
Table S2: What is CDHIT-cluster?
CD-HIT clusters described in paragraph (lines 163-170):
Clustering of 171 highly abundant and germline-specific sequences was performed using CD-HIT-EST (v4.6, with parameters: -c0.8, -G0, -aS 0.3, -aL 0.3, -sc 1, -g 1, -b 4 ) [52], resulting in the identification of 30 clusters (S2 Table). We then cross aligned (blastn with -word_size 11) [53] sequences from separate clusters and found that some clusters could be further merged (required to have at least 4 hits), resulting in 20 clusters, 8 of which contained multiple sequences and 12 of which were singletons. For characterization of larger-scale repetitive structures, genomic scaffolds with the largest number of hits to each germline-specific element were identified by BLAST alignment (blastn, -word_size 11, at least 80% of bases aligned) (S3 Table).

Round 2
Reviewer 2 Report
The PCR protocol for generating probes is vague. The extension temperature seems pretty low to me. Which type of polymerase did you use? Provide the exact protocol (how long and at what temperature each PCR step was run for each probe): either in the text or in Supp. Table 3 and 4.
Page 4, Line 159: “was” and not “were”.
Author Response
We are grateful for additional comments and corrections made by reviewer and provided necessary details to PCR labeling protocol in manuscript.
Reviewer comment
The PCR protocol for generating probes is vague. The extension temperature seems pretty low to me. Which type of polymerase did you use? Provide the exact protocol (how long and at what temperature each PCR step was run for each probe): either in the text or in Supp. Table 3 and 4.
We apologize for the confusion and thank the reviewer for noticing this. The annealing temperature was 55 oC, whereas the extension was performed at the standard 72 oC. We now include a more in-depth and accurate protocol in the Methods.
Page 4, Line 159: “was” and not “were”.
Correction has been made.
